# Using deep learning systems for diagnosing common skin lesions in sexual health
Nyi Nyi Soe [1,2] ✉, Phyu Mon Latt [1,2], David M. Lee[3], Zhen Yu[2,4], Martina Schmidt[3], Melanie Bissessor [2,3], Ei Thu Aung[2,3], Zongyuan Ge [4,5], Rashidur Rahman[3], Eric P. F. Chow[2,3,6], Jason J. Ong[2,3], Christopher K. Fairley[2,3,9] & Lei Zhang [1,2,7,8,9] ✉

## Abstract

**Background** Early identification and treatment of sexually transmitted infections (STIs) prevents complications and improves STI control. However, there are obstacles to delivering accessible care, particularly for genital conditions.

**Methods** We developed a deep learning system (DLS) using 15,891 clinical images from public repositories and the Melbourne Sexual Health Centre (MSHC) to classify 33 anogenital dermatological conditions, including STIs and non-STIs. We prospectively collected 336 images to evaluate the DLS's accuracy and compared it to the clinician diagnosis. We also evaluated whether DLS recommendations aligned with clinical urgency for seeking care based on the diagnosis.

**Results** On the hold-out test dataset, the DLS achieves an accuracy of 59.2% (top-1) (standard deviation (SD) 0.7%) and the correct diagnosis is included in the top five diagnoses (top-5) with an accuracy of 82.1% (SD 13.3%). On the 8-month prospective dataset at MSHC, the DLS achieves a top-1 accuracy of 52.1%, top-3 of 73.8%, and top-5 of 89.9%. The performance varies across 33 diagnoses, with the majority (77%) of the diagnoses achieving over 80.0% for top-5 accuracy. The DLS recommendation based on top-5 diagnoses for seeking care maintains 100% sensitivity for urgent cases (e.g. syphilis) but a lower positive predictive value (59.5%). The recommendation based on top-1 diagnosis provides more balanced sensitivity (85.0%) and PPV (80.5%).

**Conclusions** The DLS demonstrates satisfactory statistical accuracy that would have been inadequate for clinical use. Future work should evaluate the DLS's performance across expanded populations and skin conditions from multiple clinics in different countries and determine how such tools could be used for the public good.

## Plain language summary

Sexually transmitted infections (STIs) can cause serious health problems if not treated quickly. We developed a computer program that uses artificial intelligence (AI) to identify skin conditions from photos of genital areas. We trained this AI system using over 15,000 images to recognise 33 different conditions, including STIs like syphilis and herpes, and other skin conditions. We then tested it with new images from clients at Melbourne Sexual Health Centre. The AI correctly included the right diagnosis in its top 5 suggestions 90% of the time. While not accurate enough to replace doctors, this technology could help people decide whether they need urgent medical care if used as a public symptom checker.

Worldwide, more than one million new cases of sexually transmitted infections (STIs) are acquired daily[1–3] with profound implications for affected individuals and substantial societal costs[4,5]. Early detection and treatment of STIs is key to their effective control. However, services for STIs face challenges in delivering accessible healthcare as further rises in STIs put further pressure on already overwhelmed STI services. This leads to a vicious cycle, where the increasing burden of STIs overwhelms healthcare services, which itself leads to higher STI rates.

The adoption of digital health technologies could improve healthcare delivery by providing more accessible and convenient healthcare solutions.

In dermatology practice, healthcare providers adopted a teledermatology approach to improve service efficacy and expand access to specialists[6]. In this approach, the specialists evaluate digital images of skin conditions with clinical data, showing a reasonable diagnosis accuracy but lower than conventional in-clinic consultations. It also demonstrated improved satisfaction from both patients and providers[7,8]. Furthermore, healthcare providers have implemented software systems designed to identify individuals at high risk of acquiring STIs and predict potential clinical diagnoses. These systems have used various technologies, including simple rule-based decision-making and complex machine learning algorithms[9–18]. Currently, there

is no automated programme specifically designed to differentiate anogenital skin lesions related to sexual health practices based on clinical images.

One particular digital technology that has demonstrated potential beneficial health outcomes is using artificial intelligence (AI), particularly Deep Learning System (DLS), for screening and medical diagnosis[19,20]. For instance, DLS can accurately differentiate various skin lesions based on dermoscopic and clinical images[21–24]. Our previous study[25] also demonstrated that DLS can accurately distinguish mpox skin lesions from common STI skin lesions. Raquel et al. successfully employed DLS to classify four types of genital lesions, such as herpes, warts, and condylomas[26]. Moreover, Groh et al[27] demonstrated that a DLS-based decision support system trained to differentiate 46 skin diseases significantly improved the visual diagnosis accuracy of clinicians by more than 33%. Currently, no study has specifically developed and evaluated a DLS for the classification of anogenital skin diseases related to sexual health.

This study aims to develop and evaluate a DLS using clinical images to correctly predict various anogenital skin conditions. The development of the DLS could potentially assist individuals in deciding how early to seek medical attention or serve as a decision support tool for clinicians. Such a tool could also potentially improve access to healthcare services through prioritising depending on the likelihood they had an important STI, particularly in settings with limited health resources.

## Methods

This study was conducted at the Melbourne Sexual Health Centre (MSHC), which is the largest sexual health centre in Australia. The study was conducted in accordance with the ethical regulations and guidelines. Ethical review was approved by the Alfred Hospital Ethics Committee (Project Number: 524/21 and 683/22). We adhered to the reporting standards for artificial intelligence in health care, outlined in the MINimum Information for Medical AI Reporting (MINIMAR) recommendations[28].

### Data sources

We obtained clinical sexual health images from two sources[1]: publicly available online repositories and[2] an image dataset from the MSHC. For the publicly available dataset, we obtained permission to use de-identified images from public repositories (See details in data availability and Table S1). At MSHC, since January 2010, the images were taken by clinicians using compact digital cameras or, more recently, a bespoke smartphone application called *Image Capture* that was developed in-house[29]. Written informed consent was obtained from clients for the use of their images in research. All images were then de-identified. The images contained 33 types of common anogenital dermatological conditions at MSHC[1]: STIs-related conditions (e.g. genital warts, herpes simplex virus, molluscum contagiosum, mpox, syphilis and syphilis rash) and[2] non-STI conditions (e.g. pearly penile papules, balanitis, dermatosis, lichen sclerosus, non-syphilis related skin rashes and normal anatomical variants) (See details in Table S1).

### Image selection criteria

Three experienced sexual health clinicians (CKF, DL, MS) and two researchers (NS, PL) manually reviewed the images to determine if the listed diagnosis was consistent with the clinical image. For MSHC images that had not been de-identified, clinical notes and laboratory results were also reviewed in the clinic's Electronic Health Record (EHR) before a copy of the image was de-identified. For the public datasets, the clinicians verified the diagnosis visually. Any images that did not achieve a diagnostic consensus among the clinicians were excluded from our image dataset. We excluded duplicated and low-resolution images to maintain the quality and uniqueness of the dataset.

### Data splitting

We used all images from public datasets ($n = 7999$) and MSHC images collected from January 2010 to May 2023 ($n = 7892$) for DLS training and testing. We used a fivefold cross-validation method to minimise potential bias. This method split the dataset ($n = 15,891$) into training (80%) and testing (20%) datasets. We repeated this process five times, each time shuffling the dataset to ensure varied images in each fold. (See Fig. S1). The training dataset ($n = 12,713$) was used for training and internal validation, while the testing dataset ($n = 3178$) served as a hold-out test set for external validation.

### Data pre-processing

We manually cropped the images to concentrate on the areas of interest, discarding any irrelevant background to optimise the model training. We uniformly resized all the images to dimensions of $640 \times 640$ pixels. To enhance the diversity of our training dataset, we applied various data augmentation methods. These included random cropping, horizontal/vertical flipping, and random changes to brightness and contrast levels.

### Deep Learning System (DLS) training and validation

We designed a multi-classification model aimed at predicting the most probable five dermatological conditions from a given image. Our approach used a Convolutional Neural Networks (CNN) architecture. Specifically, we employed a transfer learning approach on a pre-trained YOLOv8 image classification model[30], which was fine-tuned on our dataset with common dermatological conditions observed within a sexual health clinic. An overview of the CNN architecture is presented in Fig. 1. During the training process, pre-processed images were input into convolutional layers to extract image features. The classifier layers then utilised these features to make predictions based on the images. To address the class imbalance, we used the class weight function in YOLOv8 to automatically assign higher weights to underrepresented conditions during the training. Model training was conducted on a Tesla T4 GPU machine using the Python programming language (version 3.8.2).

We conducted the training process for 100 epochs and determined the best-performing model based on the evaluation metrics of top-1 accuracy. The accuracy is defined as the proportion of correct predictions made by DLS out of all its predictions for each specific condition. Top-1 accuracy assesses whether the DLS's most confident prediction matches the actual diagnosis for each image. Top-5 accuracy assesses whether the actual diagnosis is among the DLS's top five most confident predictions for each image. We implemented five iterations of 5-fold cross-validation, calculating the metrics for each fold in each iteration. The model with the best validation performance (saved as 'best.pt' by YOLOv8 was selected during the training process, and this best-performing model was evaluated on the independent test dataset.

### DLS integration to Image Capture App

After DLS training and validation, we integrated the best-performing DLS into our existing Image Capture App within an integrated development environment (IDE). Image Capture is a web-based application utilised at MSHC to collect clinical images (Fig. 2). Eligible clients receive an SMS link granting access to the Image Capture App. They can use smartphone cameras to capture images, which are then submitted directly to a secure database. The system allows the research team to conduct a real-time assessment of the model's performance.

### DLS evaluation using a prospective dataset

We prospectively collected images ($n = 336$) with the Image Capture App and digital camera from June 2023 to January 2024 to evaluate the model performance in a clinical setting. We excluded poor-quality images (low lighting, lack of focus/clarity) and those in which the clinicians could not reach a consensus on a visual diagnosis. We compared DLS predictions to final diagnoses confirmed through laboratory testing or by expert clinical assessment. To assess the DLS's predictive accuracy, we first computed the overall top-1, top-3, and top-5 accuracy. Then, we calculated these performance metrics for each disease type and evaluated the site-specific top-5 accuracy.

We also evaluated whether the prediction from DLS would result in the same advice about attending health care as the actual diagnosis would have.

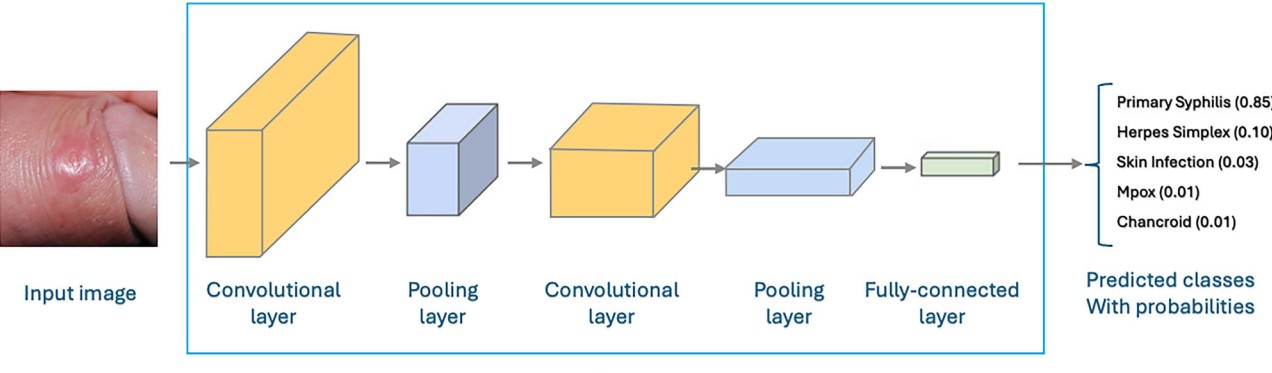

**CNN**

**Fig. 1 | Convolutional Neural Network (CNN) architecture for diagnostic prediction.** The CNN processes input clinical images through alternating convolutional layers that extract image features and pooling layers that reduce spatial dimensions. The final fully connected layer outputs probability scores for each diagnostic class via softmax activation. Example output classes include primary syphilis, herpes simplex, skin infection, mpox, and chancroid.

To do this, we used recommendations from our own website about how quickly individuals with specific symptoms should attend health care[15]. Recommendations were categorised into three intervals based on the diagnosis[1]: within 24 h[2], within the next 2–3 days, and[3] within the next 7 days (Table S2). We aimed to determine whether the DLS recommendations could potentially lead to delays in seeking clinical care compared to the clinical recommendations.

### Reporting summary
Further information on research design is available in the Nature Portfolio Reporting Summary linked to this article.

## Results
### Sample characteristics
Our study used a total of 15,891 images for both model training and validation. Genital warts (11.8%), herpes simplex virus (10.2%), dermatosis (9.1%) and lichenoid dermatosis (8.9%) were the most common disease classes in the dataset (Details in Table S1). Disease classes that contributed less than 1.0% of the dataset were folliculitis, mpox, penile papules, candidiasis, phimosis, chancroid, cystic lesions, and bacterial vaginosis. The remaining disease classes constituted between 1.0% and 6.4% of the dataset. In the prospective image dataset ($n = 336$), herpes simplex virus (17.6%), gonorrhoea/chlamydia (11.6%), lichenoid dermatosis (11.6%), primary syphilis (8.6%), genital warts (7.7%), balanitis (7.4%) and dermatosis (6.8%) were identified as the most common disease classes.

### Deep Learning System performance on the training dataset
We monitored DLS optimisation during training with top-1 and top-5 accuracy as well as training and validation loss for each training round (epoch) (Table 1). Higher top-1 and top-5 accuracy values indicate better performance across the 33 types of skin conditions and lower loss values indicate greater learning efficacy in our model. During training, the DLS reached a performance plateau after 60 epochs, demonstrating a mean top-1 accuracy of 0.663 (SD 0.012), a mean top-5 accuracy of 0.914 (SD 0.003), training and validation losses of 0.009 (SD 0.000) and 1.444 (SD 0.004), respectively (Table 1 and Fig. S2). The confusion matrix with training data (Fig. S3A) visually represents the frequency of the DLS's accurate and mistaken predictions for each condition during the training process.

### DLS performance on the testing dataset
We again evaluated the DLS on the testing dataset for each fold. The model achieved a mean top-1 accuracy of 0.592 (SD 0.007) and a mean top-5 accuracy of 0.821 (SD 0.133). The confusion matrix with the testing data (Fig. S3B) shows that most STI conditions, except condylomata lata and chancroid, achieved over 60.0% top-1 accuracy. Condylomata lata was often confused with genital warts, dermatosis, and herpes, while chancroid was frequently confused with chancre, herpes and genital warts. Non-STI classes, except drug reaction, naevus, phimosis and skin infections, achieved 60.0–83.0% top-1 accuracy. Shingles was frequently misclassified with the herpes virus and drug reaction was often misclassified as a non-syphilis rash. Additionally, naevus was also frequently confused with malignant pigmented lesions.

### DLS evaluation on the prospective dataset
We selected the optimal model for evaluation using a prospective dataset, achieving a top-1 accuracy of 0.521, top-3 accuracy of 0.738, and top-5 accuracy of 0.899. Detailed accuracy metrics for each disease class are illustrated in Fig. 3. The majority of disease classes, except drug reaction, secondary syphilis and scabies, achieved over 80.0% top-5 accuracy. Similarly, most classes achieved over 60.0% top-3 accuracy. Top-1 accuracy varies substantially between disease classes, ranging from 15.0% to 100.0%. The diseases with the highest model accuracy were dermatosis, penile discharge associated with gonorrhoea/chlamydia, molluscum contagiosum and mpox, while those with the lowest accuracy were herpes zoster, scabies and penile papules.

The relationship between training sample size and model performance was complex. While some conditions with larger training datasets performed well (e.g., genital warts, $n = 1873$, achieved 88.0% top-5 accuracy), sample size alone did not determine accuracy. The visual distinctiveness of conditions had a strong influence on model performance. Gonorrhoea/chlamydia ($n = 240$) and mpox ($n = 142$) achieved 100.0% top-5 accuracy despite smaller training sets, while scabies ($n = 212$) showed lower performance at 62.0%.

### DLS performance stratified by disease and lesion sites
We conducted the subgroup analysis for correct and incorrect predictions for each skin condition by site of the lesion (Figs. S4.1 and S4.2). We observed the imbalanced lesion sites in the dataset and the most common sites were anogenital regions, including glans penis, penile shaft and corona, perianal/buttock and anus. For example, in molluscum contagiosum and mpox, the conditions were correctly predicted across different lesion sites. However, we observed some site-specific misclassifications. For example, herpes simplex virus and chancres affecting the anus and vagina/vulva were more frequently misclassified. Similarly, scabies and folliculitis on the penile shaft and drug reactions on the penile corona were also incorrectly predicted. A few incorrect predictions occurred for genital warts in the anus and on the penile shaft. However, we cannot demonstrate a significant difference because of the insufficient sample size for each class.

### Comparing DLS and clinician recommendations for timing to visit the clinic
Confusion matrices (Fig. 4) compared the DLS recommendations with clinicians' regarding the timing for attending the clinic. The top-5 diagnoses

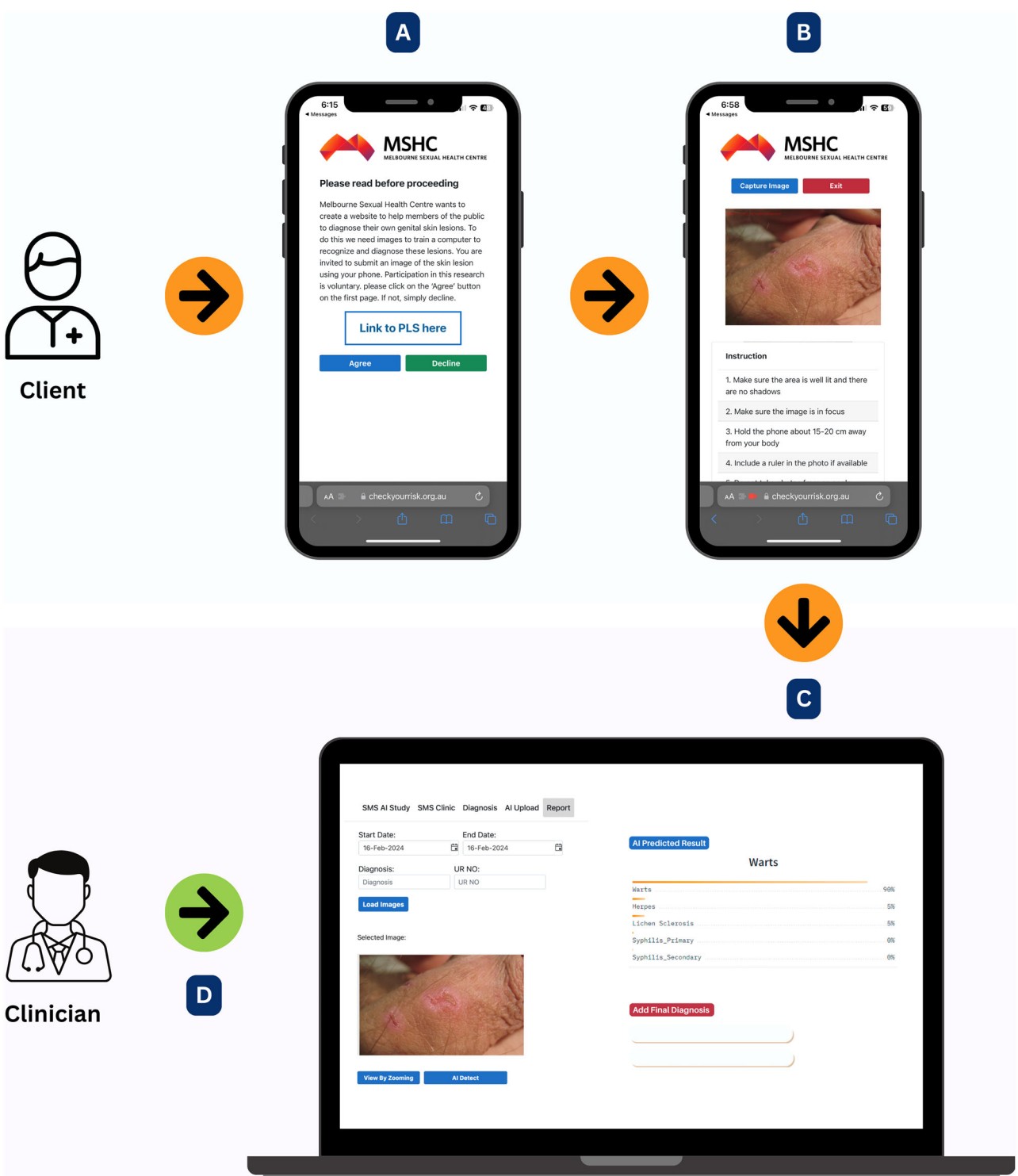

**Fig. 2 | Workflow for the Image Capture app and diagnosis verification. A** MSHC clients received a text message with a link to view the participant information statement and provide consent to participate in this study. **B** Participants could anonymously upload images of skin lesions to the MSHC secured server using their smartphone camera via Image Capture application. **C** Uploaded images were stored and accessible through the EHR system at MSHC to clinicians and researchers. **D** Experienced sexual health clinicians verified diagnoses for the uploaded images using laboratory testing or clinical assessment.

model (Fig. 4A) accurately advised clinic visits within 24 h for all urgent cases (100% sensitivity), but a lower Positive Predictive Value (PPV) of 59.5% and higher provider burden. The top-2 diagnoses model (Fig. 4C) maintained high sensitivity (95.2%) for urgent cases and improved PPV (66.5%) with a few delayed cases. The top-1 diagnosis model (Fig. 4D) had a balanced sensitivity (85%) and PPV (80.5%) and minor delays.

### Visualisation of DLS attention regions for predictions
We generated heatmaps to visualise the image regions the model attended to when predicting the possible conditions (Fig. 5). We found that the model generally focused on appropriate image regions containing the primary lesions. However, accuracy was reduced when disturbances like hair occluded the lesion site, as demonstrated in the example of misclassified genital warts.

### Table 1 | Summary of model performance on training, testing and prospective datasets

| 1. Model Performance on Training Dataset (n = 12,713) | |
|---|---|
| Top-1 Accuracy | 0.663 (SD 0.012) |
| Top-5 Accuracy | 0.914 (SD 0.003) |
| Training loss | 0.009 (SD 0.000) |
| Validation loss | 1.444 (SD 0.004) |
| **2. Model Performance on Testing Dataset (n = 3178)** | |
| Top-1 Accuracy | 0.592 (SD 0.007) |
| Top-5 Accuracy | 0.821 (SD 0.133) |
| **3. Model Performance on Prospective Dataset (n = 336)** | |
| Top-1 Accuracy | 0.521 |
| Top-3 Accuracy | 0.738 |
| Top-5 Accuracy | 0.899 |

## Discussion

This study is the first to evaluate a deep learning system for diagnosing 33 dermatological conditions within a sexual health clinic setting, using a prospective dataset for validation. Our findings suggest that the DLS can diagnose dermatological conditions with reasonable accuracy based on a single clinical image, although the results were better for some conditions (e.g., molluscum contagiosum). However, the accuracy is not sufficient to replace a clinical diagnosis based on history, examination, and where appropriate clinical investigations. In some ways, comparing the analysis of a single image to a clinical diagnosis is not a 'fair' comparison because the true diagnosis had been arrived at with the help of a clinical history and laboratory test. However, it did answer the question of whether such DLS could replace clinical care, and it could not. We also analysed if the DLS would be useful if the true diagnosis was included in the first 1, 3, or 5 differential diagnoses. This greatly improved the sensitivity for the true diagnosis to be within one of these differential diagnoses, but also substantially reduced the specificity of the DLS. In this context, we tested

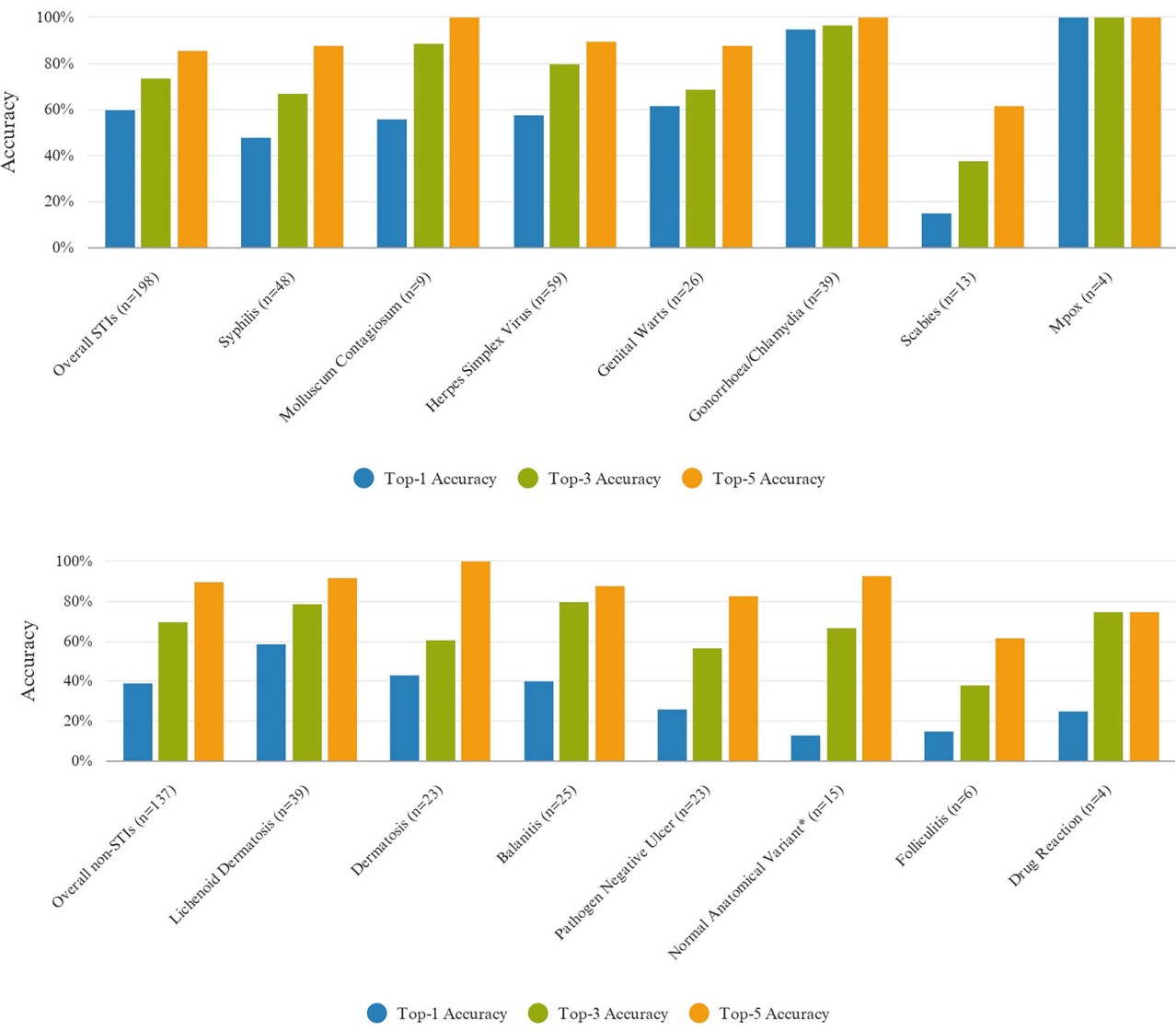

**Fig. 3 | Model accuracy per disease class on the prospective dataset.** Bar charts display DLS performance across disease categories. Top panel: Sexually transmitted infections showing overall and individual class accuracies. Bottom panel: Non-STI dermatological conditions showing overall and individual class accuracies. Accuracy is the proportion of correct prediction made by DLS out of its all predictions. Top-1 accuracy is the percentage of tested images for which the DLS correctly predicts the true dermatological condition as the most likely diagnosis. Top-3 and top-5 accuracy measure how often true condition is within the DLS's top 3 and top 5 predicted conditions. Normal anatomical variant category includes normal anatomical skin conditions, penile papules, phimosis and cystic lesion. Syphilis category includes primary syphilis (chancre), condylomata lata and secondary syphilis rash.

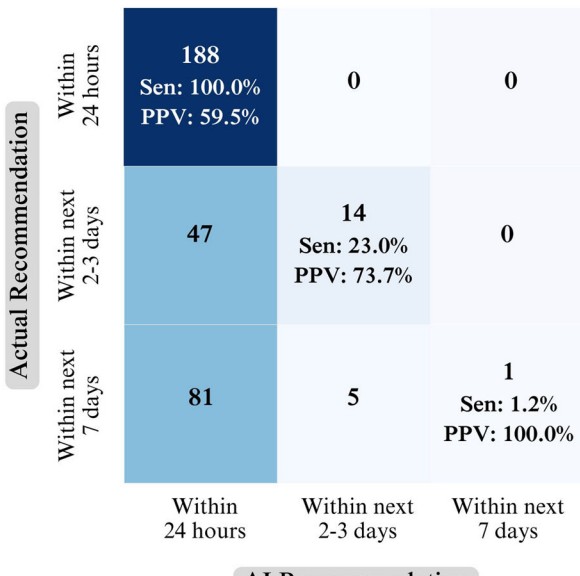

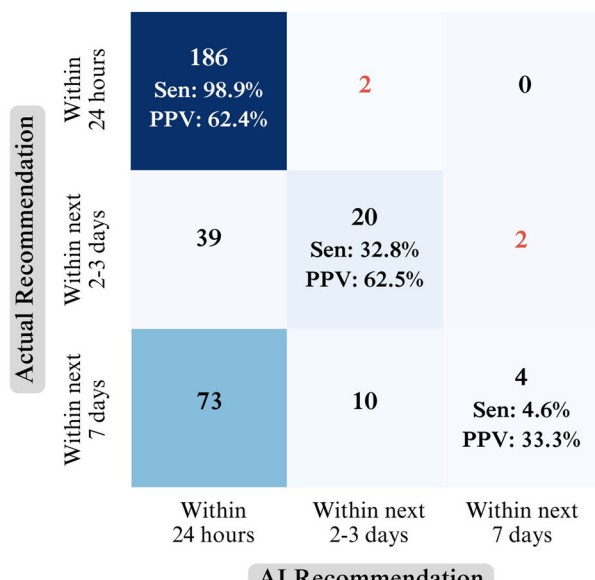

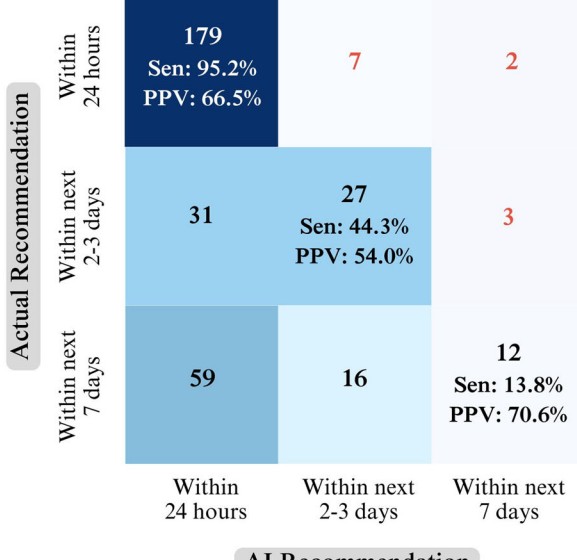

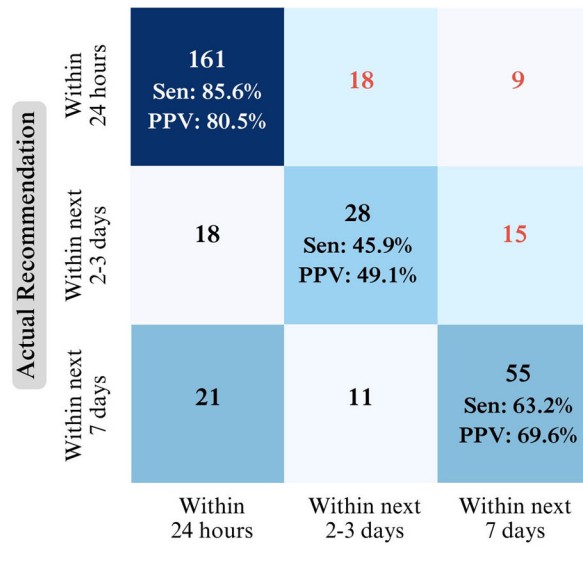

**Fig. 4 | Comparison of the recommendations of the DLS vs clinician regarding the timing to visit the clinic.** Confusion matrices compare the AI recommendation and clinician recommendation for urgency of review, stratified by number of AI diagnoses considered. **A** Top-5 AI diagnoses vs clinician recommendation. **B** Top-3 AI diagnoses vs clinician recommendation. **C** Top-2 AI diagnoses vs clinician recommendation. **D** Top-1 AI diagnosis vs clinician recommendation. Cells show the number of cases in each category and darker shading indicates higher counts. Sensitivity (Sen) and positive predictive value (PPV) of DLS are reported for each urgency level.

whether recommendations to attend health care for important diagnoses were useful if the important diagnosis was included in the top 3 or 5 diagnoses. Indeed, the sensitivity approached 100% but the positive predictive value was lower. We foresee that the next step is to determine whether incorporating the clinical history with images improves the DLS accuracy and exactly how such DLS might augment clinical care, particularly in situations where the clinicians are not Sexual Health specialists. Furthermore, it is possible that if a system is set up appropriately, it could encourage

the public with important STIs such as syphilis to attend health care urgently to reduce onward transmission of these conditions or ensure those with important diagnoses who were ambivalent about attending health care to do so.

There are no other studies that have assessed STI images primarily using multi-class classification, although there have been studies looking at more general dermatological conditions. Studies by Rafay et al.[31] and Aboulmira et al.[32] examining a similar number of dermatological

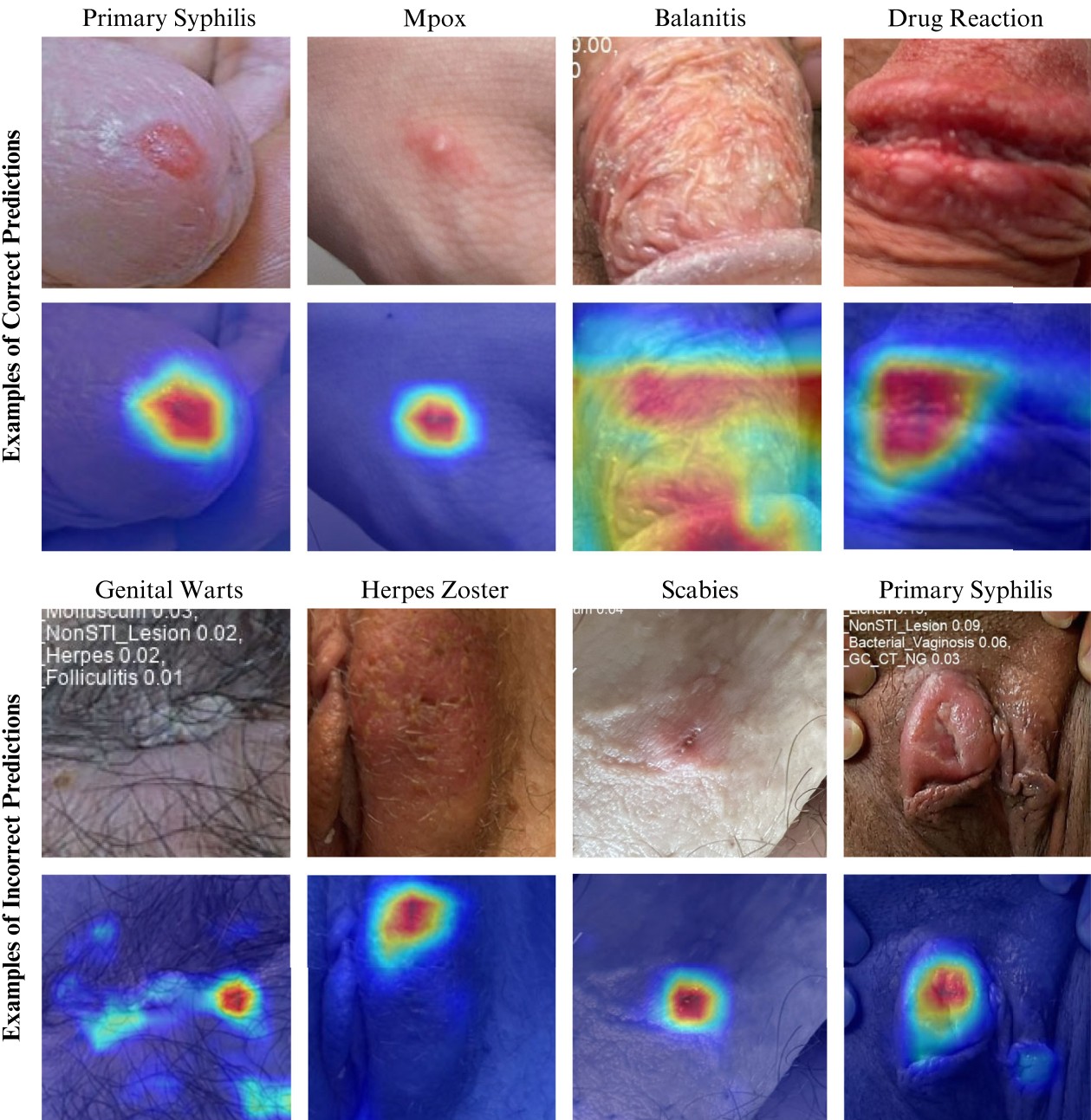

**Fig. 5 | Heatmaps visualise important regions in the image that influenced the model's prediction.** Warmer colours (red and yellow) indicate areas the model focused on when making decision. Cooler (blue) colour zones were less important.

conditions as we did and compared to our DLS they reported a higher top-1 accuracy using EfficientNet-B2 and DenseNet (87.2% and 68.9%, respectively). However, their findings are not directly comparable to ours due to the differences between our study and theirs in relation to the type of algorithm used, the quality of the images and the similarity between included dermatological conditions and lack of external validation on prospective datasets[33]. In contrast, Escalé-Besa et al.[8] conducted a prospective validation of an image model across 44 skin diseases with 100 patients, achieving a top-1 accuracy of 48.0% and a top-5 accuracy of 89.0%. Similarly, Groh et al.[27] reported a top-1 accuracy of 47.0% for their model across 44 skin diseases. These study findings align with the accuracy demonstrated by our model, further supporting the validity of our results. As multi-class classification with more diverse disease categories remains challenging, future research should investigate advanced architectures, such as Vision Transformers and Vision Language Models, to improve diagnostic accuracy.

The comparison of our results, which rely on only a single image, to expert clinicians who assess patient information and investigation results, highlights the limitations of our approach. This may explain why our DLS failed to predict correctly for certain dermatological conditions, especially for those with similar presentations that often require laboratory testing to accurately diagnose (e.g., herpes simplex vs. herpes zoster, herpes ulcer vs. chancre). While AI diagnosis could incorporate parts of the clinical history, it will not be able to access laboratory results. Additionally, clinicians can visually inspect skin lesions from multiple angles and distances during an in-person assessment and our DLS uses a single image. From a sexual health perspective, while misclassifying some non-STI lesions as STI lesions and prompting individuals to seek health care may be acceptable, incorrectly misclassifying STI lesions as non-STIs would place individuals at risk from complications but also allow onward transmission to sexual partners. The inclusion of clinical metadata, such as patient demographics and clinical history, into the analysis of images could

potentially optimise the performance of DLS and therefore enhance its translational utility. Recent studies[34–36] demonstrated that integrating additional clinical metadata improved the DLS's overall accuracy by 2.9% to 5% in general dermatological and STI images, compared to the DLS using images alone.

One potential way that our DLS could be used is in the healthcare setting to assist clinicians in making a diagnosis or suggesting the correct laboratory investigations, such as including tests for herpes zoster with a clinical lesion consistent with herpes simplex. Further research should explore the DLS's potential to be used as a diagnostic decision-support tool for assisting healthcare providers with varying levels of sexual health expertise, ranging from sexual health specialists to general practitioners. Groh et al[27] demonstrated that DLS decision support significantly improved the diagnostic accuracy of both specialists and generalists by over 33% for general dermatological conditions. However, no study has yet investigated the impact on clinician's decision-making specifically within sexual healthcare.

Another potential use of our DLS would be to make it public-facing and investigate whether it resulted in an earlier presentation to health care for those with STIs. A qualitative study by King et al.[37] found such an AI application for the identification of anogenital skin lesions is generally acceptable and has potential to increase access to care and provide early insights into STI-related conditions. This could promote individual's self-care for STIs, enabling them to recognise their symptoms and seek timely medical attention, thereby reducing ongoing transmission risk. Online access for checking symptoms could be particularly useful for those living long distances from clinics if the lesions could be identified as not requiring health care[38,39]. However, we must carefully address the potential risks and benefits before implementation, avoiding unintended consequences while communicating AI predictions to the community[40]. It would also be important to balance the need to encourage testing without causing undue concern, given the sensitive nature of an STI diagnosis[41,42]. In this context, future studies should explore the optimal way of communicating the results to the public.

Our study has several limitations. First, most of the images we used for DLS training and testing were previously collected images in a single sexual health clinic, and these images may differ in their quality from images that are collected in general practice or in dermatological clinics. Studies would need to assess this limitation by using images collected in different clinical environments. It is useful to know that our DLS achieved consistent performance in evaluating an 8-month prospective dataset despite the training occurring on previously collected images. Second, the ground truth diagnosis of the clinician may differ between STIs and non-STIs, as STI diagnoses are often confirmed by laboratory testing, while non-STIs rely more on clinical evaluation. Moreover, we cannot confirm whether pigmented lesions from public databases were histologically verified, as the source repositories did not provide this information. This could lead to potential bias in the ground truth diagnosis used to train the DLS for certain disease types. Our study did not provide a benchmark for clinicians' visual diagnosis accuracy to compare with DLS performance, as we evaluated the DLS against final diagnoses confirmed by clinicians' complete assessment and laboratory tests. Third, although our DLS was trained and tested on 33 diseases, some uncommon conditions were not included in the prospective dataset used for model evaluation because they were not present in the prospective data collection period. Fourth, we did not incorporate the clinical data related to the lesions (e.g., duration, presence of pain) due to data anonymisation and different skin tones. Instead, we conducted subgroup analyses based on lesion sites and visualised DLS attention using heatmaps during prediction. However, our study did not analyse model performance across different skin tones, which may impact the generalisability of our findings across diverse populations. Future studies should prioritise the evaluation of model performance by skin tone. Fifth, our disease categorisation was designed for practical model training and sexual health triage rather than following traditional dermatological

classification. Some diseases were grouped into a group of conditions based on image availability and clinical urgency in sexual health, which may not align with standard dermatological terminology. Finally, the DLS does not automatically detect the abnormal areas within the entire image. Users need to manually crop the image, focusing on the main lesion areas to define the 'region of interest' for the DLS. The quality and preciseness of the cropping could affect the DLS prediction. Cropping is also limited when multiple lesions occur at the same anatomical site. Future research should examine whether integrating the current model with an autodetection system for skin lesions could potentially enhance the model prediction accuracy.

## Conclusion

Our study on external validation of DLS with a prospective dataset demonstrated promising consistency in the performance in differentiating across 33 skin conditions. Our DLS performed a similar level of accuracy as achieved during initial DLS development and testing, indicating a strong potential for utilising DLS in sexual health settings. Additional research should aim to further validate model performance across expanded patient populations and skin conditions from multiple clinics in different countries.

## Data availability

The images of the datasets are available from the following databases. The Danderm dataset is available from https://danderm-pdv.is.kkh.dk/atlas/index.html. The Kaggle skin lesion dataset can be assessed at https://www.kaggle.com/datasets/shubhamgoel27/dermnet. The Fitz17k is available on GitHub (github.com/mattgroh/fitzpatrick17k). Part of the MSHC dataset is available at https://stiatlas.org/. Source data for Fig. 3 can be found in Supplementary Data 1.

## Code availability

We used YOLOv8 image classification models from Ultralytics. The codes for model training, validation and evaluation can be found at https://docs.ultralytics.com/tasks/classify/.

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

## Acknowledgements

The authors thank Monash University for a PhD scholarship for author N.S. We extend our gratitude to the clinicians at MSHC who contributed to this study. We acknowledge the open-source Danderm, Kaggle and Fitz17k dermatology image repositories that provided datasets integral to this study. C.K.F. is supported by an NHMRC Leadership Investigator Grant (GNT1172900). J.O. is supported by the NHMRC investigator grant (GNT1193955). E.P.F.C. is supported by an NHMRC Leadership Investigator Grant (GNT2033299), and E.T.A. is supported by EPFC's NHMRC Leadership Investigator Grant (GNT2033299).

## Author contributions

Conceptualisation: L.Z., C.K.F., N.S., E.P.C.F.; Study design: L.Z., C.K.F., N.S., P.L.; Image collection and verification: N.S., C.K.F., D.L., J.O., M.S., M.B., E.A.; Software development: N.S., Z.Y., R.R., Z.G.; Data curation: N.S., P.L.; Formal analysis: N.S., P.L., E.P.F.C.; Writing: N.S., P.L., C.K.F., L.Z.; Review and editing manuscript: all authors reviewed and approved the final manuscript.

## Competing interests

The authors declare no competing interests.

## Additional information

[1]Artificial Intelligence and Modelling in Epidemiology Program, Melbourne Sexual Health Centre, Alfred Health, Melbourne, VIC, Australia. [2]School of Translational Medicine, Faculty of Medicine, Nursing and Health Sciences, Monash University, Melbourne, VIC, Australia. [3]Melbourne Sexual Health Centre, Alfred Health, Melbourne, VIC, Australia. [4]AIM for Health Lab, Faculty of Information Technology, Monash University, Melbourne, VIC, Australia. [5]Department of Data Science and AI, Faculty of Information Technology, Monash University, Melbourne, VIC, Australia. [6]Centre for Epidemiology and Biostatistics, Melbourne School of Population and Global Health, The University of Melbourne, Melbourne, VIC, Australia. [7]China-Australia Joint Research Centre for Infectious Diseases, School of Public Health, Xi'an Jiaotong University Health Science Centre, Xi'an, Shaanxi, PR China. [8]Phase I clinical trial research ward, The Second Affiliated Hospital of Xi'an Jiaotong University, Xi'an, Shaanxi, PR China. [9]These authors contributed equally: Christopher K. Fairley, Lei Zhang. ✉e-mail: drnyinyisoe1989@gmail.com; lei.zhang1@monash.edu

