## [Transparent Peer Review file · Communications Medicine]

Using Deep Learning Systems for Diagnosing Common Skin Lesions in Sexual Health

Corresponding Author: Dr Nyi Nyi Soe

Version 0:

Reviewer comments:

Reviewer #2

(Remarks to the Author)

The manuscript presents the development and evaluation of a DLS for classifying 33 dermatological conditions of the anogenital area. The DLS achieved a top-1 accuracy of 59.2% and 52.1%, and top-5 accuracy of 82.1% and 89.9%, respectively. The manuscript highlights the potential use of AI tools to assist clinical triage and support diagnosis in sexual health settings.

Overall impression of the work: This study addresses an underexplored area at the intersection of artificial intelligence and sexual health. The manuscript is well written and structured. I would suggest some methodological clarifications and improvements to fully support the conclusions and to enhance the model's reproducibility and potential translational use.

Specific comments:

1. The dataset includes many underrepresented conditions and there is no mention of how class imbalance was addressed during training. Please specify whether techniques were applied. If not, consider discussing the implications of imbalance on model performance, especially for rare but clinically significant conditions.
2. The model uses image data exclusively, without incorporating key clinical data, such as symptom duration or anatomical site. I would suggest to discuss the potential benefit of metadata inclusion, and consider referencing recent studies showing performance improvement through multimodal input.
3. There is a typographical error in Figure 2: "cliniician" should be corrected to "clinician".
4. No benchmark is provided comparing the DLS performance to that of clinicians, either overall or for difficult cases. Including even a limited benchmark from the prospective dataset would contextualize the DLS accuracy.

Overall, this is a valuable and timely contribution that could inform future developments in AI-assisted dermatological diagnosis, particularly in the field of sexual health.

Reviewer #3

(Remarks to the Author)

This is an interesting study and it is helpful to report negative results.

There needs to be better categories for the conditions - these are confusing

Methods - data sources paragraph

Pearly penile papules are a normal variant but seems to be a separate category

'Non-syphilis related skin rashes' is far too vague. NB. The term 'rash' only should be used - there are no other types of rash apart from those affecting the skin so 'skin rash' is unnecessary

Lichen sclerosis is spelt incorrectly - it should be lichen sclerosus

Results p.7

What are you including in lichenoid dermatosis?? If you are including lichen simplex chronicus and lichen sclerosus, then

these are NOT lichenoid dermatoses in the proper meaning of the term - ie those with a lichenoid histological pattern, so this should really only include lichen planus - but then you need to stipulate what type of LP - classic, erosive, hypertrophic??

Page 18

Again this is very confusing - what is a dermatosis - these would include eczema and psoriasis but they are separate Non-STI lesions?? again too vague

Were pigmented lesions histologically confirmed?

The times to report symptoms and be seen are completely unworkable and incorrect in clinical practice. To recommend that a patient with VIN is seen within 24 hours is really unnecessary - this can be present for years and the risk of invasive cancer is small.

Reviewer #5

(Remarks to the Author)

Thank you for inviting me to review this paper, reporting on the development and evaluation of a CNN-architecture for diagnosing common sexual health conditions based on images of skin lesions. Overall, the paper is well written and clearly reported, and refreshingly, the authors provide a balanced assessment of the clinical implications of their findings, in particular, that it is not ready for clinical use yet. However, the relatively low performance does not undermine the quality of the study and the authors have made feasible suggestions for further improvements and testing. In particular, the analyses of recommendations on time to visit are informative with regards to practical use, and the heatmaps of the model attention provide useful insights into model learning. I have a few specific comments:

1. To what extent are the conditions on which the model performs better, those that are more common in the data, e.g. scabies has relatively few cases overall. Can you correlate accuracy with numbers in the dataset?
2. Methods, lines 157-158: I was unclear on how the best-performing model was selected. Was a model outputted for each epoch and then one selected, or only at the end of 100 epochs? Presumably top-1 and top-5 accuracy may vary, in which case, which was preferred? And was this the accuracy on the training or testing datasets?
3. Methods/discussion: with image based learning and skin lesions, skin colour may bias learning. You may not have the data on skin colour or ethnic background to report, but this is worth discussion as an important area for future investigation.
4. Discussion, limitations: are there any more recent model architectures that might perform better, and that could be compared in future?
5. Discussion: lower performance may be more acceptable if it still performs better than clinicians, particularly non-specialists. Is any further research planned, for example, comparing evaluation between specialists, non-specialists and the model?
6. Two small typos: Figure 2: "Clinician" and line 238 – 'except' rather than expect

Version 1:

Reviewer comments:

Reviewer #2

(Remarks to the Author)

I believe the current version is ready for publication and does not require further changes.

Reviewer #3

(Remarks to the Author)

Reviewer #5

(Remarks to the Author)

Thank you to the authors for addressing all of my comments - I have no further comments.

Reviewer Comments:

Reviewer #1:

Comment: The manuscript presents the development and evaluation of a DLS for classifying 33 dermatological conditions of the anogenital area. The DLS achieved a top-1 accuracy of 59.2% and 52.1%, and top-5 accuracy of 82.1% and 89.9%, respectively. The manuscript highlights the potential use of AI tools to assist clinical triage and support diagnosis in sexual health settings.

Overall impression of the work: This study addresses an underexplored area at the intersection of artificial intelligence and sexual health. The manuscript is well written and structured. I would suggest some methodological clarifications and improvements to fully support the conclusions and to enhance the model's reproducibility and potential translational use.

Response: Thank you for your thoughtful and constructive feedback. We have carefully addressed your points to improve the manuscript.

Specific comments:

Comment: 1. The dataset includes many underrepresented conditions and there is no mention of how class imbalance was addressed during training. Please specify whether techniques were applied. If not, consider discussing the implications of imbalance on model performance, especially for rare but clinically significant conditions.

Response: Thanks for raising this important point about class imbalance. We acknowledge that our dataset includes underrepresented conditions. We addressed this class imbalance by data augmentation, as mentioned in the data-processing section on page 4, lines 144-146.

“To enhance the diversity of our training dataset, we applied various data augmentation methods. These included random cropping, horizontal/vertical flipping, and random changes to brightness and contrast levels.”

We also addressed this class imbalance by applying a “class weighting” function during the model training. This is a built-in function in YOLOv8 that addresses class imbalance through its loss calculation. The text is added to the “Deep Learning System Training and Validation” section, on page 5, lines 155-157.

“To address the class imbalance, we used the class weight function in the YOLOv8 to automatically assign higher weights to underrepresented conditions during the training.”

Comment: 2. The model uses image data exclusively, without incorporating key clinical data, such as symptom duration or anatomical site. I would suggest to discuss the potential benefit of metadata inclusion, and consider referencing recent studies showing performance improvement through multimodal input.

Response: Thank you for this valuable suggestion. We agree that including metadata is crucial in AI applications, especially in clinical settings. In fact, we have previously compared model performance with and without metadata in our earlier work¹ (available at <https://rdcu.be/eoAvh>). The reason we did not include metadata in the current study is that we aimed to use a larger, more diverse dataset combining images from multiple sources (public repositories and MSHC), and not all images had associated clinical metadata available. We discussed this important consideration in our manuscript (page 12, lines 375-379).

“The inclusion of clinical metadata such as patient demographics and clinical history into the analysis of images could potentially optimise the performance of DLS and therefore enhance its translational utility. Recent studies (33-35) demonstrated integrating additional clinical metadata improved the DLS’s overall accuracy by 2.9% to 5% in general dermatological and STI images, compared to the DLS using images alone.”

1. Soe, N.N., Yu, Z., Latt, P.M. et al. Evaluation of artificial intelligence-powered screening for sexually transmitted infections-related skin lesions using clinical images and metadata. BMC Med 22, 296 (2024). <https://doi.org/10.1186/s12916-024-03512-x>

Comment: 3. There is a typographical error in Figure 2: “cliniician” should be corrected to “clinician”.

Response: We have corrected the typo, 'cliniician' to 'clinician' in Figure 2 (Page 6).

Comment: 4. No benchmark is provided comparing the DLS performance to that of clinicians, either overall or for difficult cases. Including even a limited benchmark from the prospective dataset would contextualize the DLS accuracy.

Response: Our study did not directly compare DLS performance with clinicians' visual diagnostic accuracy. Instead, we used the final diagnosis (confirmed by experienced clinicians through comprehensive assessment and laboratory testing) as the ground truth to evaluate DLS accuracy. We agree that a benchmark comparison would provide important context for interpreting our results. We will include such comparisons in our future studies to compare the DLS performance with the visual diagnosis accuracy of clinicians, where we will evaluate how the DLS performs relative to clinicians making diagnoses based solely on visual assessment alone. We have discussed this in our manuscript. (page 13, lines 383-385 and 413-415)

“Further research should explore the DLS’s potential to be used as a diagnostic decision-support tool for assisting healthcare providers with varying levels of sexual health expertise, ranging from sexual health specialists to general practitioners.”

“Our study did not provide a benchmark for clinicians’ visual diagnosis accuracy to compare with DLS performance, as we evaluated the DLS against final diagnoses confirmed by clinicians’ complete assessment and laboratory tests.”

Comment: Overall, this is a valuable and timely contribution that could inform future developments in AI-assisted dermatological diagnosis, particularly in the field of sexual health.

Response: We sincerely thank you for your thoughtful review and constructive feedback. Your comments have helped us significantly improve the quality of our manuscript.

Reviewer #2 :

Comment: This is an interesting study and it is helpful to report negative results.

There needs to be better categories for the conditions - these are confusing

Response: Thanks for your suggestions and comments on the disease categorisation. We acknowledge that our categorisation was based on prioritising the most important STIs and the availability of sufficient images per disease category, rather than a strict dermatological classification. This does not affect our recommendations for healthcare-seeking in sexual health.

We appreciate your very insightful suggestion for disease categorisation. We will consider more standardised dermatological categories after consulting with the dermatologists in future studies. We truly appreciate your valuable comments, which will help us further improve our research.

Comment: Methods - data sources paragraph

Pearly penile papules are a normal variant but seems to be a separate category

'Non-syphilis related skin rashes' is far too vague. NB. The term 'rash' only should be used - there are no other types of rash apart from those affecting the skin so 'skin rash' is unnecessary

Lichen sclerosis is spelt incorrectly - it should be lichen sclerosUs

Response:

Pearly penile papules: We agree that these are normal variants. We kept them as a separate category because we had enough images (n = 71) for model training. Other normal variants with fewer images were grouped into another category.

Non-syphilis-related skin rash: Syphilis is an important disease in sexual health. We included this group to differentiate between 'secondary syphilis rash' and other types of rash.

Lichen sclerosus: We have corrected the typo, 'lichen sclerosis' to 'lichen sclerosus' (page 4, line 121).

Comment: Results p.7

What are you including in lichenoid dermatosis?? If you are including lichen simplex chronicus and lichen sclerosus, then these are NOT lichenoid dermatoses in the proper meaning of the term - ie those with a lichenoid histological pattern, so this should really only include lichen planus - but then you need to stipulate what type of LP - classic, erosive, hypertrophic??

Response: Thanks again for explaining this. We acknowledge that our use of this term deviates from the strict dermatological definition. In our dataset, “lichenoid dermatosis” included lichen sclerosus and lichen planus, without consideration of the specific types based on histological pattern. The images in this category came from both public dermatology repositories (Danderm, Kaggle, Fitz17k) and our Melbourne Sexual Health Centre database. The purpose of including different varieties is to improve the AI model’s ability to differentiate STIs from these lichenoid conditions. We acknowledge this deviation from standard dermatological terminology as a limitation in our manuscript (page 13, lines 423-426). We will consider consulting with the dermatologists for the terminology in future studies.

“Fifth, our disease categorisation was designed for practical model training and sexual health triage rather than following traditional dermatological classification. Some diseases were grouped into a group of conditions based on image availability and clinical urgency in sexual health, which may not align with standard dermatological terminology.”

Comment: Page 18

Again this is very confusing - what is a dermatosis - these would include eczema and psoriasis but they are separate

Non-STI lesions?? again too vague

Response: We acknowledge this deviation from standard dermatological terminology as a limitation in our manuscript (page 13, lines 423-426). We included eczema (n=484) and psoriasis (n=1,024) as separate categories because there were sufficient images for model training. We will consider combining them in our future studies. For non-STI lesions, the lesions were confirmed with laboratory tests and no pathogen was detected. We were unable to identify the diagnosis further, and consequently, we classified them as non-STI lesions. The aim of including them is for AI to learn their features and differentiate STIs from these non-STI conditions.

“Fifth, our disease categorisation was designed for practical model training and sexual health triage rather than following traditional dermatological classification. Some diseases were grouped into a group

of conditions based on image availability and clinical urgency in sexual health, which may not align with standard dermatological terminology.”

Comment: Were pigmented lesions histologically confirmed?

Response: Thank you for this important question. The pigmented lesion images were obtained from established dermatology databases commonly used for AI research, as mentioned in our data availability section (Page 15, lines 467-471). While these image databases provide expert-annotated diagnoses, they do not specify whether pigmented lesions were histologically confirmed. We acknowledge this as a limitation in our manuscript (Page 13, lines 410-412).

“Moreover, we cannot confirm whether pigmented lesions from public databases were histologically verified, as the source repositories did not provide this information. This could lead to potential bias in the ground truth diagnosis used to train the DLS for certain disease types.”

Comment: The times to report symptoms and be seen are completely unworkable and incorrect in clinical practice. To recommend that a patient with VIN is seen within 24 hours is really unnecessary - this can be present for years and the risk of invasive cancer is small.

Response: We acknowledge that some recommended timeframes may appear overly conservative. Our recommendations were based on the guidelines from our current practice (<https://www.staystifree.org.au/how-urgent-are-my-symptoms>) and input from sexual health physicians. We agree that VIN specifically may not require 24-hour urgency given its typically slow progression. However, in sexual health settings, we encourage prompt evaluation of anogenital lesions primarily to prevent STI transmission and ensure early treatment of infectious conditions. We appreciate your insight and will refine these timeframes prior to implementing the AI model in clinical practice.

Reviewer #3:

Comment: Thank you for inviting me to review this paper, reporting on the development and evaluation of a CNN-architecture for diagnosing common sexual health conditions

based on images of skin lesions. Overall, the paper is well written and clearly reported, and refreshingly, the authors provide a balanced assessment of the clinical implications of their findings, in particular, that it is not ready for clinical use yet. However, the relatively low performance does not undermine the quality of the study and the authors have made feasible suggestions for further improvements and testing. In particular, the analyses of recommendations on time to visit are informative with regards to practical use, and the heatmaps of the model attention provide useful insights into model learning. I have a few specific comments:

Response: Thank you for your positive feedback and valuable suggestions to improve our manuscript. We have carefully considered your comments and revised the manuscript.

Comment: 1. To what extent are the conditions on which the model performs better, those that are more common in the data, e.g. scabies has relatively few cases overall. Can you correlate accuracy with numbers in the dataset?

Response: Thank you for this insightful question about the correlation between sample size and model performance. While larger training datasets generally improve performance, we found that visual distinctiveness of conditions had a stronger influence on accuracy in our study. We have added the text to the result section (page 8, lines 250-255).

“The relationship between training sample size and model performance was complex. While some conditions with larger training datasets performed well (e.g., genital warts, n=1,873, achieved 88.0% top-5 accuracy), sample size alone did not determine accuracy. The visual distinctiveness of conditions had a strong influence on model performance. Gonorrhoea/ chlamydia (n=240) and mpox (n=142) achieved 100% top-5 accuracy despite smaller training sets, while scabies (n=212) showed lower performance at 62.0%.”

We also discussed the model confusion among diseases with similar appearances on Page 7, lines 234-239.

“Condylomata lata was often confused with genital warts, dermatosis, and herpes, while chancroid was frequently confused with chancre, herpes and genital warts. Non-STI classes, except drug reaction, naevus, phimosis and skin infections, achieved 60.0-83.0% top-1 accuracy. Shingles was frequently misclassified with the herpes virus and drug reaction was often misclassified as a non-syphilis rash. Additionally, naevus was also frequently confused with malignant pigmented lesions.”

Comment: 2. Methods, lines 157-158: I was unclear on how the best-performing model was selected. Was a model outputted for each epoch and then one selected, or only at the end of 100 epochs? Presumably top-1 and top-5 accuracy may vary, in which case, which was preferred? And was this the accuracy on the training or testing datasets?

Response: We prefer the top-1 accuracy to choose the best model based on the top-1 accuracy during the model training, and we have revised the text accordingly. We also provided the performance for training, testing and prospective datasets in Table 1.

During training, YOLOv8 automatically saves two models: 'best.pt' (the model with the best validation performance) and 'last.pt' (the model from the final epoch). We selected the 'best.pt' model, which represents the checkpoint with the highest validation performance across all epochs. We have revised the text on page 5, lines 169-171.

“The model with the best validation performance (saved as ‘best.pt’ by YOLOv8 was selected during the training process, and this best-performing model was evaluated on the independent test dataset.”

Comment: 3. Methods/discussion: with image based learning and skin lesions, skin colour may bias learning. You may not have the data on skin colour or ethnic background to report, but this is worth discussion as an important area for future investigation.

Response: We acknowledge that we did not analyse the model performance by skin color, which is an important limitation. We have added this as a limitation in our manuscript (page 13, lines 420-422).

“However, our study did not analyse model performance across different skin tones, which may impact the generalisability of our findings across diverse populations. Future studies should prioritise the evaluation of model performance by skin tone.”

Comment: 4. Discussion, limitations: are there any more recent model architectures that might perform better, and that could be compared in future?

Response: We have added the discussion for exploring more recent model architectures in the future study in our manuscript (Page 12, lines 361-363).

“As multi-class classification with more diverse disease categories remains challenging, future research should investigate advanced

architectures, such as Vision Transformers and Vision Language Models (VLMs), to improve diagnostic accuracy.”

Comment: 5. Discussion: lower performance may be more acceptable if it still performs better than clinicians, particularly non-specialists. Is any further research planned, for example, comparing evaluation between specialists, non-specialists and the model?

Response: Thanks for this helpful suggestion. We will consider conducting further research to evaluate the diagnosis accuracy of the model with specialists and non-specialists in sexual health. We also discussed this on page 13, lines 383-385 and 413-415.

“Further research should explore the DLS’s potential to be used as a diagnostic decision-support tool for assisting healthcare providers with varying levels of sexual health expertise, ranging from sexual health specialists to general practitioners.”

“Our study did not provide a benchmark for clinicians’ visual diagnosis accuracy to compare with DLS performance, as we evaluated the DLS against final diagnoses confirmed by clinicians’ complete assessment and laboratory tests.”

Comment: 6. Two small typos: Figure 2: “Cliniician” and line 238 – ‘except’ rather than expect

Response: Thank you for pointing out these typos. We have corrected 'expect' to 'except' in line 238 and 'cliniician' to 'clinician' in Figure 2.